# A Self-Training Method for Semi-Supervised GANs

## Abstract

Since the creation of Generative Adversarial Networks (GANs), much work has been done to improve their training stability, their generated image quality, their range of application but nearly none of them explored their self-training potential. Self-training has been used before the advent of deep learning in order to allow training on limited labelled training data and has shown impressive results in semi-supervised learning. In this work, we combine these two ideas and make GANs self-trainable for semi-supervised learning tasks by exploiting their infinite data generation potential. Results show that using even the simplest form of self-training yields an improvement. We also show results for a more complex self-training scheme that performs at least as well as the basic self-training scheme but with significantly less data augmentation.

## 1 Introduction

Generative Adversarial Networks (GANs) have shown impressive results in a variety of image synthesis tasks (Denton et al., 2015; Radford et al., 2015; Im et al., 2016; Yoo et al., 2016). Their adversarial training has also been used to make supervised learning more robust (Salimans et al., 2016). However, not much work has been done to use the generated data as a source of knowledge. The use of generated data offers a new dimension to data augmentation and model training (Patki et al., 2016).

This is especially useful when data is limited and costly as in the semi-supervised learning paradigm. This paradigm has known much success allowing systems with much less labelled data and much more unlabelled data to perform nearly as well as if the data were all labelled.

In this work, we propose a self-training meta-learning method that can be applied to GANs that make use of their generated data in order to increase their classification performance. We base ourselves on the Improved GAN (Salimans et al., 2016) since it already has mechanisms to cope with semi-supervised learning. We compare a simple baseline self-training method with a more advanced scheme inspired from Self-training with selection-by-rejection by Zhou et al. (2012).

In Section 2, we first present related work done in the areas of semi-supervised learning and of self-training. In Section 3, we then present the basic theory behind GANs, the Improved GAN and the self-training method upon which we base our work. In Section 4, we present the two self-training algorithms we compare. In Section 5, we present the results of our self-training experiments compared with the vanilla Improved GAN and we finally conclude in Section 6.

## 2 Related Work

The idea of semi-supervised learning is not new: as labelled training data can be expensive to obtain, semi-supervised learning (Zhu, 2005; Chapelle et al., 2009) becomes an important topic when one has only access to a small amount of labelled training data and a large amount of unlabelled training data. Works regarding semi-supervised learning have been previously done for word sense disambiguation (Yarowsky, 1995; Abney, 2004), for Gaussian random fields (Zhu et al., 2003) and for learning word representation (Turian et al., 2010).

When paired with deep learning, semi-supervised learning has had a few success stories from Cat-GAN (Springenberg, 2015), from Sutskever et al. (2015) introducing a new cost function, from the Improved GAN by Salimans et al. (2016) and from Triple GAN by Li et al. (2017). However, most of the work in GANs has been focused on improving the visual quality of the generated images (Denton et al., 2015; Radford et al., 2015; Im et al., 2016; Warde-Farley & Bengio, 2016; Yoo et al., 2016; Arjovsky & Bottou, 2017) and their training stability (Salimans et al., 2016; Arjovsky & Bottou, 2017) while not much work has been done on improving the GAN's performance using its own generated data. The work we base ourselves on, Improved GAN (Salimans et al., 2016), will be explained later with the Technical Background Section 3.

On the self-training side, before the advent of deep learning, Hearst (1991) and Yarowsky (1995) used self-training for word sense disambiguation, Riloff et al. (1999) used self-training in the form of bootstrapping for information extraction and later for learning subjective nouns (Riloff et al., 2003) with Nigam et al. (2000) using EM for text classification. More recently, self-training has been used for object recognition (Rosenberg et al., 2005; Zhou et al., 2012). These techniques allowed semi-supervised learning models to train themselves through multiple rounds while seeing their performance increase as the rounds go.

## 3 TECHNICAL BACKGROUND

GANs are composed of two agents: a discriminator $\mathcal{D}$ and a generator $\mathcal{G}$. They are typically both deep neural networks with each their set of parameters. The goal of $\mathcal{D}$ is to distinguish between real images and generated images coming from $\mathcal{G}$. Each is trained successively through many rounds to reach an equilibrium where $\mathcal{G}$ produces images indistinguishable from real images and where $\mathcal{D}$ can only randomly guess if the image is real or generated. The minimax objective they are trying to optimize is

$$\min_{\mathcal{G}} \max_{\mathcal{D}} \mathbb{E}_{x \sim p_{\text{data}}(x)} \left[\log \mathcal{D}(x)\right] + \mathbb{E}_{z \sim p_z(z)} \left[\log(1 - \mathcal{D}(\mathcal{G}(z)))\right]$$

where $p_{\text{data}}$ is the real data distributionm, where $p_z$ is a noise distribution to be used by the generator and where $\mathcal{D}(x)$ must be as close to 1 as possible when $x$ is a real image.

The Improved GAN brings a modification to the traditional objective to allow semi-supervised training with the use of the generated data for classification tasks. First, the discriminator $\mathcal{D}$ the authors use is a classifier. Thus, instead of outputting a single number representing the probability of a true example, $\mathcal{D}$ outputs a softmaxed vector for each classes. Additionally, they introduce a fake class for the generated images, the $K + 1$th class. As a result, $\mathcal{D}$ must be able to predict the class $x$ belongs to, including the "generated" class. The new loss function for $\mathcal{D}$ is thus

$$\mathcal{L} = \mathcal{L}_{\text{supervised}} + \mathcal{L}_{\text{unsupervised}}$$

where

$$\mathcal{L}_{\text{supervised}} = -\mathbb{E}_{x,y \sim p_{\text{data}}(x,y)} \log p_{\text{model}}(y \mid x, y < K + 1)$$

$$\mathcal{L}_{\text{unsupervised}} = -\{\mathbb{E}_{x \sim p_{\text{data}}(x)} \log \left[1 - p_{\text{model}}(y = K + 1 \mid x)\right] + \mathbb{E}_{x \sim \mathcal{G}} \log \left[p_{\text{model}}(y = K + 1 \mid x)\right]\}$$

and where $p_{\text{model}}$ is the predicted probability of the discriminator $\mathcal{D}$.

For the generator $\mathcal{G}$, the authors introduce the *feature matching* loss which goes as follows:

$$\|\mathbb{E}_{x \sim p_{\text{data}}} f(x) - \mathbb{E}_{z \sim p_z} f(G(z))\|_2^2$$

where $f(x)$ is the activation of an intermediate layer of $\mathcal{D}$ when given the input $x$. This should prevent mode collapse and allow for a more stable training for the GAN.

We also take inspiration from the self-training method Self-training with selection-by-rejection proposed by Zhou et al. (2012). They aim to add unlabelled data in which the classifier is confident and which are influential to its decision boundary.

They start with a labelled dataset $L$ and an unlabelled dataset $U$ and train classifier $h$ on $L$. Samples far from $h$'s decision boundary are considered ones where $h$ is confident about their labels. The authors use the negative entropy as a proxy to the distance of an example to the decision boundary. Now associate each sample in $U$ with a weight that will be used for sampling and label the samples in $U$ with $h$. Take the half of $U$ with samples furthest to the decision boundary to be candidates

for addition to the labelled dataset, call this set $U_\delta$. From $U_\delta$, subsets $U_i \subseteq U_\delta$ are randomly sampled according to the weights attributed to each sample and each of those subsets $U_i$ have a corrupted counterpart $U_i'$ where the samples are identical to those in $U_i$ except for their labels which are randomly changed to another. The idea is that the set $U_i$ which is the most influential to the decision boundary will yield a hypothesis $h_i$ trained on $L \cup (U \setminus U_i) \cup U_i'$ that disagrees the most in predictions with a hypothesis $\hat{h}$ which is simply trained on $L \cup U$. This set is added to $L$ and the weights associated with those added samples are decayed. We note that because of the weighted sampling, examples that were previously added can be re-added with another label in further rounds of self-training.

## 4 METHODOLOGY

In this work, we compare two self-training schemes with the vanilla Improved GAN which does not use self-training. This section details the different algorithms we use.

The first self-training scheme is a basic one which simply adds unlabelled data according to some confidence threshold. This Basic Self-Training algorithm is detailed in Algorithm 1.

The second is a scheme nearly identical to the Self-Training through Selection-by-Rejection explained in Section 3 with a few distinctions. The first distinction is that we substitute the weighted sampling of $U_i$ with a fixed sized uniform random sampling of $U_\delta$. We also keep track of the unlabelled examples that were added to the labelled dataset to make sure that they are never added again in further self-training rounds. Instead, we allow the main hypothesis to relabel all added examples after every self-training round. We have found that these modifications work better in the GAN case with the addition of generated data. This Improved Self-Training algorithm is detailed in Algorithm 2.

The network architectures used in this work are the same as those used in the original Improved GAN paper and the feature matching loss is used for the generator.

**Input** :
- Labelled dataset L
- Unlabelled dataset U
- Confidence threshold threshold
- Number of self-training rounds num_rounds

**Output:** Self-Trained GAN via Basic Self-Training Scheme

```
1  function BasicSelfTrain(L, U):
2      Û ← U \L
3      for iter = 1 to num_rounds do
4          h ← new GAN trained on L and U
5          for x in Û do
6              P(y|x) ← h.predict (x)
7              if max P(y|x) > threshold then
8                  L ← L ∪ {(x, argmaxP(y|x))}
9                  Û ← Û \{x}
10         end
11         relabel all added data from the beginning in L
12         gen_data ← h.generate()
13         U ← U ∪ gen_data
14         Û ← Û ∪ gen_data
15     end
16     h ← new GAN trained on L and U
17  return best trained GAN
```

**Algorithm 1:** Basic Self-Training Algorithm

**Input :**
- Labelled dataset $\mathsf{L}$
- Unlabelled dataset $\mathsf{U}$
- Number of subsets $\mathsf{n}$
- $\mathsf{U_r}$ sample fraction sample_frac
- Number of self-training rounds num_rounds

**Output:** Self-trained GAN

1 **function** CalculateDisagreement($\mathsf{h_1}$, $\mathsf{h_2}$, $\mathsf{S}$):
2      disagreement $\leftarrow \|\mathsf{S}\| - \|\{x \in \mathsf{S} \mid \mathsf{h_1}.\mathsf{predict(x)}$ is equal to $\mathsf{h_2}.\mathsf{predict(x)}\}\|$
3      **return** disagreement/$\|\mathsf{S}\|$

4 **function** SelfTrain($\mathsf{L}$, $\mathsf{U}$):
5      $\hat{\mathsf{U}} \leftarrow \mathsf{U} \setminus \mathsf{L}$
6      **for** $iter = 1$ *to* num_rounds **do**
7          $\mathsf{h} \leftarrow$ new GAN trained on $\mathsf{L}$ and $\mathsf{U}$
8          $\hat{\mathbf{h}} \leftarrow$ new GAN trained on $\mathsf{L}$ and $\mathsf{U}$ labelled by $\mathsf{h}$
9          **foreach** $\mathsf{x} \in \hat{\mathsf{U}}$ **do** $\mathsf{d[x]} \leftarrow \sum\limits_{\text{labels } \mathsf{l}} \mathsf{P(l|x)} \log \mathsf{P(l|x)}$

10
11          $\delta \leftarrow \mathtt{median}(\mathsf{d})$
12          $\mathsf{U_\delta} \leftarrow \{\mathsf{x} \in \hat{\mathsf{U}}$ such that $\mathsf{d[x]} > \delta\}$
13          **for** $i = 1$ to $\mathsf{n}$ **do**
14              $\mathsf{U_i} \leftarrow$ randomly sample sample_frac of $\mathsf{U_\delta}$
15              $\mathsf{U_i'} \leftarrow$ randomly change the labels associated with examples in $\mathsf{U_i}$ as labelled by $\mathsf{h}$
16              $\mathbf{h_i} \leftarrow$ new GAN trained on $\mathsf{L} \cup \mathsf{U_i'} \cup (\mathsf{U} \setminus \mathsf{U_i'})$ labelled by $\mathsf{h}$
17          **end**
18          $\mathsf{U_r} \leftarrow \underset{\mathsf{U_i}}{\arg\max}$ CalculateDisagreement($\hat{\mathbf{h}}$, $\mathbf{h_i}$, $\mathsf{U}$)
19          relabel all added data from the beginning in $\mathsf{L}$
20          $\mathsf{L} \leftarrow \mathsf{L} \cup \mathsf{U_r}$
21          gen_data $\leftarrow \mathsf{h}.\mathsf{generate()}$
22          $\mathsf{U} \leftarrow \mathsf{U} \cup$ gen_data
23          $\hat{\mathsf{U}} \leftarrow (\hat{\mathsf{U}} \setminus \mathsf{U_r}) \cup$ gen_data
24      **end**
25      $\mathsf{h} \leftarrow$ new GAN trained on $\mathsf{L}$ and $\mathsf{U}$
26 **return** *best trained GAN*

**Algorithm 2:** Improved Self-Training Algorithm

The running time for one round of self-training is much higher in the improved self-training scheme. The most significant part is the training of the hypotheses $\hat{h}, h_1, \cdots, h_n$ which are trained for the same number of epochs as the main GAN.

**Dataset** Following the evaluation scheme from Improved GAN (Salimans et al., 2016), our methods are tested against the MNIST dataset of handwritten digits (Lecun et al., 1998) and CIFAR-10 (Krizhevsky & Hinton, 2009). MNIST contains a total of $80\,000$ labelled images of size 32 by 32 pixels, black and white where $60\,000$ of them are reserved as the training set and $20\,000$ are for the testing set. CIFAR-10 is a dataset of natural coloured images also of size 32 by 32 pixels with $50\,000$ train images and $10\,000$ test images. The experiments presented in the later Section 5 on Experimental Results had GANs trained on the whole training datasets and the test results were on the test sets. During the experiments, a subset of the training data has been kept labelled to be taken as the labelled dataset while the rest had their labels removed for the unsupervised part of the training. The hyperparameter search was done on a validation set that was about a third of MNIST's training set when there were 10 labelled examples for each of the 10 classes making 100 labelled examples in total.

**Hyperparameters** In both of these self-training algorithms, we have found experimentally that retraining the classifier from scratch at the beginning of every self-training iteration yielded better results than continuing the training with the added data. Following this, the hypotheses $\hat{h}$ and $h_i$ in our version of the self-training through rejection are also retrained from scratch at every round. The number of rounds to run the self-training can be tuned but good improvements are already observed after 2 or 3 rounds. The confidence threshold for addition of data to the labelled training set in the basic self-training scheme was set to 0.95, i.e. unlabelled examples $x$ were added if $\max P(y \mid x) > 0.95$, with $P(y \mid x)$ being the softmaxed output of the classifier. This results in a large proportion of data being added even on the first self-training round but we found that setting this threshold higher, e.g. to the median or to the mean of $\max P(y \mid x)$ over all $x$, lead to a worse performance. For the other self-training scheme, the number of subsets $U_i$ to draw from $U_\delta$ has been set to 4. While the authors claim that having a higher number of sets is better to select more permutations of data points it increases the computation time. To make each $U_i$, we randomly sampled one fifth of $U_\delta$. In other words, one fifth of $U_\delta$ corresponding to $U_i$ is corrupted and the rest is uncorrupted when training $h_i$. The disagreement calculation between hypotheses was done on the whole unlabelled dataset rather than only on a select subset of data as in the original paper. This is a choice we made with hopes that having more data to compute disagreement on would lead to a better estimate of what affects the decision boundary. An exhaustive search of hyperparameters was not performed but we have tested a few combinations through a held-out validation set and found that these parameters worked well.

## 5 EXPERIMENTAL RESULTS

The self-training algorithms were tested against two datasets: MNIST and CIFAR-10. For MNIST, the GANs were trained for 550 epochs everytime they required training and the results for the self-training schemes were after 2 rounds of self-training iterations, i.e. data augmentation was performed twice so the GANs were retrained three times in total. For CIFAR-10, each GAN was trained for 300 epochs and also for 2 rounds of self-training. The `count` parameter indicates how many samples per class are labelled in MNIST while the count remained at 400 for CIFAR-10. Each MNIST experiment was done over three seeds and the results were averaged; the results for CIFAR-10 are averaged over two seeds. The error bounds correspond to one standard deviation. MNIST results can be seen in Table 1 and CIFAR-10 results can be seen in Table 2.

We notice that having some kind of self-training yields better results than no self-training at all. Most importantly, this improvement was observed after only 1 or 2 rounds of data augmentation. The results shown in the tables are the best ones after 2 rounds, but in some cases, the best results were obtained immediately after 1 round.

In the case of MNIST, We observe that the basic self-training scheme performs well but we must keep in mind that in the basic self-training scheme, a significant amount of data is added at every round. Indeed, for the counts of 10 and 20, over $98\%$ of the unlabelled examples was added (about $59\,000$) while for the more complex self-training scheme, only one fifth of the unlabelled data is

| Counts per class | 5 | 10 | 20 |
|---|---|---|---|
| | *Best Error Rates* | | |
| **Vanilla** | $0.1359 \pm 0.1295$ | $0.0085 \pm 0.0003$ | $0.0102 \pm 0.0011$ |
| **Basic Self-Training** | $0.1019 \pm 0.1255$ | $0.0080 \pm 0.0003$ | $0.0098 \pm 0.0013$ |
| **Improved Self-Training** | $0.1201 \pm 0.1202$ | $0.0081 \pm 0.0001$ | $0.0097 \pm 0.0009$ |
| | *Improvements over Vanilla* | | |
| **Vanilla** | $0$ | $0$ | $0$ |
| **Basic Self-Training** | $0.0340 \pm 0.0245$ | $0.0005 \pm 0.0002$ | $0.0004 \pm 0.0003$ |
| **Improved Self-Training** | $0.0158 \pm 0.0103$ | $0.0004 \pm 0.0003$ | $0.0004 \pm 0.0005$ |

Table 1: Experimental results for the error rates (lower is better) and relative improvements over the vanilla GAN training without self-training (higher is better) for different counts of labelled data: 5, 10 and 20 labelled examples per class for each of the first, second and third columns respectively. All experiments are averaged over 3 different seeds and the mean is shown with the error bounds being one standard deviation.

| | Best Error Rates | Improvements over Vanilla |
|---|---|---|
| **Vanilla** | $0.2513 \pm 0.0037$ | $0$ |
| **Basic Self-Training** | $0.2471 \pm 0.0002$ | $0.0042 \pm 0.0039$ |
| **Improved Self-Training** | $0.2231 \pm 0.0029$ | $0.0282 \pm 0.0008$ |

Table 2: Experimental results for the error rates (lower is better) and relative improvements over the vanilla GAN for CIFAR-10 (higher is better) for a count of 400. All experiments averaged over 2 different seeds and the mean is shown with the error bounds being one standard deviation.

added (around $5\,000$) but this still yields a similar result to the basic self-training scheme. When the basic self-training scheme adds a little less data, e.g. about $40\,000$, no improvement is seen. We think that this is due to the addition of data poor in information or even contradictory data compared to the improved self-training case, where fewer data yielded in an improvement. This shows that adding based on a confidence score is not enough and that we also need more information. Thus the examples that were added in this more complex schemes were informative, as informative as adding nearly all of the unlabelled training set when using the basic self-training scheme. In all cases, the added unlabelled data were assigned the correct label more than $97\%$ of the time.

## 6 CONCLUSION

This paper proposed a self-training meta-learning scheme that makes use of the generated data from GANs in order to improve the classification accuracy in the MNIST and the CIFAR-10 datasets. We noticed important improvements even after two rounds of self-training and even more so in the CIFAR-10 dataset. Our improved method of data augmentation performs similarly to the basic scheme although it uses much less labelled data. On the other hand, it is much more computationally costly. When one is looking for a quick gain in accuracy, the basic self-training scheme can be enough but when looking for more, the improved method should be used as it adds less data at each round but does a more thorough analysis of what data is best to add.

Future steps for our self-training algorithm include a more thorough theoretical analysis of the self-training and trying different knobs in the algorithm. For example, the label inversion scheme can be changed to corrupt the label to the least likely label instead of randomly corrupting it. Preliminary analyses seem to indicate that this performs worst than the random scheme but it might be worth exploring more to see the reason. The calculation of the disagreement can also be changed to the mean squared error of the prediction matrices instead of simply counting the number of differing predictions. This mean squared error might help in quantifying the disagreement. The amount of corrupted data fed to the $h_i$ can also be changed and its effect studied. The selection of $U_\delta$ was done through calculating the negative entropy which should be a proxy for the distance to the boundary but maybe other meaures can be used for the initial selection of $U_\delta$. One should also look for ways to decrease the computationl cost of the improved method; the key as we have seen is to try different subsets and to check their effect on the decision boundary. Maybe another similar testing diffrent subsets can be used without having to retrain the whole GANs.

All in all, different choices can affect the performance of self-training but our initial analysis shows that we have been successful in implementing a self-training method to GANs making use of their generated data improving the performance on semi-supervised learning tasks.

ACKNOWLEDGMENTS

We would like to thank the National Natural Science Foundation of China for previously supporting the authors to prepare for the knowledge and skills demanded by this work.

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
