# OpenReview forum: "A Self-Training Method for Semi-Supervised GANs"
_ICLR.cc/2018/Conference — Reject_

### Official Review · AnonReviewer2 · 2017-11-27
**This paper presents a straight-forward application of existing self-training approach to GAN. Although the proposed approaches are sound, the technical contribution of this paper is low, and the experiments are weak.**

**Rating:** 3
**Confidence:** 5

**Review:**

This paper proposes to use self-training strategies for using unlabeled data in GAN. Experiments on only one data set, i.e., MNIST, are conducted

Pros:
* Studying how to use unlabeled data to improve performance of GAN is of technical importance. The use of the self-training in GAN for exploiting unlabeled data is sound.

Cons:
* The novelty and technical contribution is low. The unlabeled data are exploited by off-the-shelf self-training strategies, where the base learner is fixed to GAN. Using GAN does not make the self-training strategy special to the existing self-training approaches. Thus, the proposed approaches are actually a straight application of the existing techniques. In fact, It would be more interesting if the unlabeled data could be employed to the “G” and “A” in GAN.

* In each self-training iteration, GAN needs to be retrained, whose computational cost is high..

* Only one data set is used in the experiment. Some widely-used datasets, like SVHN or CIFAR-10, are not used in the experiment.

* Important baseline methods are missing. The proposed methods should be evaluated with the state-of-the-art semi-supervised deep learning methods, such as those mentioned in related work section.

---

> ### Author Response · Authors · 2017-12-20
> **Simple does not necessarily mean low novelty**
>
> Regarding the Cons:
>
> * One important part of the paper is the use of generated data. Furthermore, while it might seem like a straightforward approach, there is no real guarantee that these methods would work as well unless we have empirical proofs. The second method, in particular, is a complex one that might go wrong in the settings of GAN.
>
> * There is indeed a computational cost but training many times is already something that is commonly done e.g. when searching for hyperparameters. Also, with the advance of computer hardware and parallel processing machine learning libraries like Chainer, this problem will become less important. However, decreasing the computational time while keeping the same performance is something that can be investigated in future work.
>
> * We agree, we have added results for CIFAR-10.

---

### Official Review · AnonReviewer3 · 2017-11-28
**The paper presents to combine self-learning and GAN. The basic idea is to first use GAN to generate data, and then infer the pseudo label, and finally use the pseudo labeled data to enhance the learning process. Experiments are conducted on one image data set. The paper contains several deficiencies.**

**Rating:** 4
**Confidence:** 4

**Review:**

The paper presents to combine self-learning and GAN. The basic idea is to first use GAN to generate data, and then infer the pseudo label, and finally use the pseudo labeled data to enhance the learning process. Experiments are conducted on one image data set. The paper contains several deficiencies.

1.	The experiment is weak. Firstly, only one data set is employed for evaluation, which is hard to justify the applicability of the proposed approach. Secondly, the compared methods are too few and do not include many state-of-the-art SSL methods like graph-based approaches. Thirdly, in these cases, the results in table 1 contain evident redundancy. Fourthly, the performance improvement over compared method is not significant and the result is based on 3 splits of data set, which is obviously not convincing and involves large variance.
2.	The paper claims that ‘when paired with deep, semi-supervised learning has had a few success’. I do not agree with such a claim. There are many success SSL deep learning studies on embedding. They are not included in the discussions.
3.	The layout of the paper could be improved. For example, there are too many empty spaces in the paper.
4.	Overall technically the proposed approach is a bit straightforward and does not bring too much novelty.
5.	The format of references is not consistent. For example, some conference has short name, while some does not have.

---

> ### Author Response · Authors · 2017-12-20
> **The new experiments on CIFAR-10 should add power to the experiments**
>
> 1. Firstly, yes, we agree and we have added results for CIFAR-10, see above. Secondly, what we wanted to show was the success of self-training on the Improved GAN which already does some semi-supervised learning. Thirdly and fourthly, the results might seem similar for both self-training methods but they still show an improvement over a non-self-trained GAN which is one of the goals of our paper. The difference is more important with the CIFAR-10 results.
> 2. We did not say "has had few success", we said "has had a few success". The former means that there was little success while the latter means that there were successes, which is what we claim. If the question is on the choice of the word "few" vs "many", then okay we can change "few" to "many".
> 3. Okay, we will rearrange the layout.
> 4. The Basic Self-Training scheme might seem obvious and straightforward but the second self-training method should not be considered obvious: label inversion, disagreement calculation and multiple subset candidates is not necessarily something that anyone can think about on top of their head. Furthermore, theoretical justifications exist in the original published paper.
> 5. Okay, we will fix the references.

---

### Official Review · AnonReviewer1 · 2017-11-30
**Interesting idea but limited novelty and impact**

**Rating:** 3
**Confidence:** 4

**Review:**

This paper presents a self-training scheme for GANs and tests it on image (NIST) data.

Self-training is a well-known and usually effective way to learn models in a semi-supervised setting. It makes a lot of sense to try this with GANs, which have also been shown to help train Deep Learning methods.

The novelty seems quite limited, as both components (GANs and self-training) are well-known and their combination, given the context, is a fairly obvious baseline. The small changes described in Section 4 are not especially motivated and seem rather minor. [btw you have a repeated sentence at the end of that section]

Experiments are also quite limited. An obvious baseline would be to try self-training on a non-GAN model, in order to determine the influence of both components on the performance. Results seem quite inconclusive: the variances are so large that all method perform essentially equivalently. On the other hand, starting with 10 labelled examples seems to work marginally better than 20. This is a bit weird and would justify at least a mention, and idealy some investigation.

In summary, both novelty and impact seem limited. The idea makes a lot of sense though, so it would be great to expand on these preliminary results and explore the use of GANs in semi-supervised learning in a more thorough manner.

[Response read -- thanks]

---

> ### Author Response · Authors · 2017-12-20
> **The second method might not be as obvious as it seems**
>
> Although it might make a lot of sense, it seems that no paper has been published about the combination of both. Many things seem obvious in hindsight but a priori we cannot know for sure if these things will work out or not.
>
> The Basic Self-Training method may seem obvious but the Improved Self-Training method is not obvious. Thinking about selecting multiple subsets of unlabelled data and inverting their labels to check their effect on the decision boundary is not something that people would immediately think about. In fact, it is not even obvious why this might work. The original paper testing this method presented theoretical justification as to why it is a good idea to try on simple hypotheses but not on a complex one such as GANs.
>
> One of the goals of the paper was to test the self-training method specifically on GANs to see how it can be applied to them. Using a baseline that is not a GAN does not seem to provide information towards that goal. The MNIST results appear equivalent in both self-training methods but they still show an improvement over a non-self-trained GAN which is one of the goals we wanted to achieve with this paper.
>
> Although one method seems obvious to use, no other publications seem to have done a similar proof of concept. Moreover, the second self-training method used should not be thought of as obvious; label inversion, disagreement calculation and multiple subset candidates is not necessarily something that anyone can think about on top of their head.

---

### Author Response · Authors · 2017-12-04
**CIFAR-10 Experimental Results**

Here are some results on the CIFAR-10 dataset.

== Error Rates ==
Vanilla Improved GAN:    0.2513 ± 0.0037
Basic Self-Training:           0.2471 ± 0.0002
Improved Self-Training:   0.2231 ± 0.0029

== Improvements Over Vanilla Improved GAN ==
Vanilla Improved GAN:    0
Basic Self-Training:           0.0042 ± 0.0039
Improved Self-Training:   0.0282 ± 0.0008

The results are averaged over 2 different seeds and the error margins represent one standard deviation.

Here, we notice significant improvement of our Improved Self-Training method over the original vanilla Improved GAN.

---

### Decision · Program_Chairs · 2018-01-29
**ICLR 2018 Conference Acceptance Decision**

**Decision:**

Reject

**Comment:**

The paper presents self-training scheme for GANs. The proposed idea is simple but reasonable, and the experimental results show promise for MNIST and CIFAR10. However, the novelty of the proposed method seems relatively small and experimental results lack comparison against other stronger baselines (e.g., state-of-the-art semi-supervised methods). Presentation needs to be improved. More comprehensive experiments on other datasets would also strengthen the future version of the paper.